# Molecular and Genetic Immune Biomarkers of Primary and Immune-Therapy Induced Hypophysitis: From Laboratories to the Clinical Practice

**DOI:** 10.3390/jpm11101026

**Published:** 2021-10-15

**Authors:** Sabrina Chiloiro, Filippo Russo, Tommaso Tartaglione, Ettore Domenico Capoluongo

**Affiliations:** 1Department of Translational Medicine and Surgery, Università Cattolica del Sacro Cuore, 00168 Roma, Italy; 2UOC Endocrinology and Diabetology, Fondazione Policlinico Universitario A. Gemelli IRCCS, 00168 Roma, Italy; 3Department of Molecular Medicine and Medical Biotechnology, Federico II University-CEINGE, 80126 Naples, Italy; russof@ceinge.unina.it; 4Department of Radiological and Haematological Sciences, Università Cattolica del Sacro Cuore, 00168 Roma, Italy; tommaso.tartaglione@unicatt.it; 5Department of Radiology and Diagnostic Imaging, Istituto Dermopatico dell’Immacolata, IDI-IRCCS, 00167 Roma, Italy; 6Azienda Ospedaliera per L’Emergenza, Cannizzaro, 95126 Catania, Italy; capoluongo@ceinge.unina.it

**Keywords:** HLA, MHC, CTLA-4, PD-1, cytokines, hypopituitarism, immune-check point inhibitors, hypoadrenalism, anti-pituitary antibodies, nivolumab, ipilimumab, diabetes insipidus

## Abstract

Hypophysitis is a rare and potentially life-threatening disease, characterized by an elevated risk of complications, such as the occurrence of acute central hypoadrenalism, persistent hypopituitarism, or the extension of the inflammatory process to the neighboring neurological structures. In recent years, a large number of cases has been described. The diagnosis of hypophysitis is complex because it is based on clinical and radiological criteria. Due to this, the integration of molecular and genetic biomarkers can help physicians in the diagnosis of hypophysitis and play a role in predicting disease outcome. In this paper, we review current knowledge about molecular and genetic biomarkers of hypophysitis with the aim of suggesting a possible integration of these biomarkers in clinical practice.

## 1. Introduction

Hypophysitis is a rare inflammatory disorder of the pituitary gland and represents an emerging disease. In recent years, an increased number of cases have been reported due to research into the etiological cause of hypopituitarism, the improvement of radiological techniques, and the use in clinical practice of drugs that may favor the occurrence of immune-related adverse events [1]. The diagnosis of hypophysitis still represents a challenge, as the differential diagnosis from focal pituitary lesions is often difficult and hypophysitis symptoms typically have a latent onset. According to the inflammatory etiology of hypophysitis, the disease symptoms may be attributed to the dysfunction of the synthesis and secretion of pituitary tropins (thyroid stimulating, follicle-stimulating, luteinizing, adrenocorticotropic and growth hormones) or the compression of the neighboring structures, such as the optical chiasma, the dura mater and the nerves of the medial wall of the cavernous sinus. The identification of the typical radiological findings of hypophysitis and the exclusion of focal pituitary lesions play a crucial role in the diagnosis [2]. However, to distinguish hypophysitis from other lesions of the pituitary gland and stalk (non-secreting and prolactin-secreting pituitary adenomas, germinomas, meningiomas, gliomas, and pituitary or infundibular metastasis) is of a crucial importance for the different therapeutic management of these disorders [3]. Surgery represents the first line of treatment for several pituitary lesions [4]; instead, however, hypophysitis may be treated with immunosuppressive therapies (such as glucocorticoids, azathioprine, mycophenolate-mofetil, rituximab) or may be conservatively managed with a clinical and radiological periodical follow-up [5]. At this specific moment, the pathological identification of an immune infiltrate within the pituitary gland represents the “gold standard” for reaching a specific diagnosis of hypophysitis. However, a pituitary biopsy is not routinely performed for potentially related complications, such as the further deterioration of pituitary function. Subsequently, this procedure remains limited to patients with an uncertain diagnosis and to patients with an indication for neuro-surgical debulking [6]. For these reasons, univocal clinical and radiological criteria for the diagnosis of hypophysitis are advocated from scientific societies as consensus or guidelines. Therefore, the identification of the molecular biomarkers of hypophysitis may help physicians to reach a definitive diagnosis.

The great heterogeneity in the etiology of hypophysitis may explain the variability of its biomarkers (Table 1), as described in recent research. These biomarkers are identified as primary or autoimmune (PAH) in cases of the primary involvement of the pituitary gland and after the exclusion of all the secondary forms [1]. Instead, secondary hypophysitis may occur in patients affected by autoimmune systemic disease or by focal cystic or neoplastic pituitary lesions. Infective hypophysitis is also described in patients with congenital or acquired immune-deficit [1]. In more recent years, hypophysitis has been recognized as a clinically significant endocrine toxicity in patients receiving treatment with immune-check point inhibitors (ICIs). ICIs act by promoting immune tumor surveillance and inhibiting immune tolerance. ICIs are monoclonal antibodies that target and inhibit either cytotoxic T lymphocyte antigen 4 (anti-CTLA4 as ipilimumab), programmed cell death-1 (anti-PD-1, as pembrolizumab, nivolumab), or its ligand PD-L1 (anti-PD-L1, as atezolizumab, avelumab and durvalumab), potentiating anti-tumor immune responses. ICIs are approved for the treatment of several malignancies, such as metastatic melanoma, non-small-cell lung cancer, renal cell carcinoma, and head and neck cancers [7]. In this review, we aim to summarize the latest research on the humoral and cell-mediated immune response and genetics of hypophysitis, and to suggest a possible integration of these biomarkers in clinical practice.

## 2. Pathogenesis of Primary Autoimmune Hypophysitis

The pathogenesis of hypophysitis has not been completely clarified, despite primary hypophysitis being recognized as an autoimmune disease [8]. The pituitary, like other endocrine glands, is very susceptible to autoimmune and inflammatory injuries, being a highly vascularized peripheral organ outside the blood–brain barrier, with both arterial and venous blood supplies and with a venous portal system [9]. The endothelial cells surround the pituitary sinusoids and form a small barrier to the passage of secreted proteins from the endocrine cells to the bloodstream, which drain rapidly through the cavernous sinus to the jugular veins. Moreover, the releasing hormones are secreted from the hypothalamic nerve terminals into capillaries in the external zone of the median eminence, are collected and delivered via the hypophysial portal veins into the anterior pituitary gland, and can act as antigens of hypophysitis.

Most of the evidence on hypophysitis derives from studies conducted on experimental autoimmune hypophysitis, which is induced through the immunization of female mice with extracts of whole pituitary gland (a process called SJL/J mouse) [10]. Through these animal models, the authors showed that hypophysitis is a biphasic disorder with an acute and a subsequent chronic phase [11]. In the acute phase of the disease, the pituitary gland is enlarged by the presence of the intraglandular immune infiltration of T- and B-lymphocytes and plasma cells [11]. However, in the chronic phase of the disease, the fibrotic process induces progressive and irreversible pituitary atrophy [11]. Interestingly, in SJL/J mouse models, the chronic phase of the disease is characterized by the reduction of the plasmatic concentration of anti-pituitary antibodies (APA) and of pituitary hormones [11].

The natural history of hypophysitis is similar in humans: the chronic phase of the disease is characterized by the clinical picture of pituitary atrophy, with a subsequent secondary empty sella syndrome [12].

## 3. The Immune Response in Primary- and Immunotherapy-Induced Hypophysitis

A large number of studies in the last 30 years were designed to investigate the immune response in primary and immunotherapy-induced hypophysitis (IIH). The recognized immune biomarkers involved in these disorders are summarized in Table 2.

### 3.1. Antibodies in Primary Autoimmune Hypophysitis

Several studies were conducted on the anti-pituitary antibodies (APA). APAs have been recognized for several years as the only molecular biomarkers for hypophysitis and were investigated with different techniques, such as the complement consumption test, immunoblotting with homogenate of human autopsy pituitaries, radioligand binding assays, and immunofluorescence [13,14]. Over the years, several attempts were made to optimize the immunofluorescence method, specifically to identify the best substrate. Experiments were conducted with pituitary slides from several animals: rats, rabbits, mice, baboons, and, eventually, humans [15]. The baboon pituitary was considered the best substrate for APA identification. The serum APAs bind to the corresponding antigens present on the pituitary sections. The antigen-antibody complexes are detected by means of a goat anti-human IgG conjugated with a fluorescein isothiocyanate (FITC) [3]. IgG FITC was adsorbed with monkey serum to remove non-specific fluorescence [3]. The sera of patients were considered positive for a APAs starting at the dilution rate of 1:8 [3]. The samples were considered positive in cases with a diffuse immunofluorescence pattern and an intracytoplasmic staining in the majority of the fields. In each assay, a positive and negative control needs to be included [3]. The clinical relevance of APAs has been keenly discussed in previous research and APAs were widely considered a pathogenic marker of hypophysitis rather than a diagnostic tool. In fact, APAs were reported in other autoimmune disorders of the pituitary gland or in autoimmune systemic diseases, such as Sheehan’s syndrome, idiopathic growth hormone (GH) deficiency, idiopathic hyperprolactinemia, idiopathic hypopituitarism, brain traumatic injury, autoimmune polyendocrine syndromes and empty sella syndrome, but also in patients with pituitary adenomas or in healthy individuals [14,16,17]. The experimental hypophysitis of SJL/J models showed that APAs may be detected with a higher concentration in the initial days after mouse immunization and gradually reduce thereafter [11]. For these reasons, the APAs were also considered clinically helpful for the diagnosis of acute hypophysitis in humans, but only if detected at a high concentration [16]. Recently, we proved that APAs are more prevalent in patients affected by PAH (68.4%) than in patients affected by not-secreting pituitary adenomas (22%) and in health controls (14%) [18]. In the same study, we found that positivity for anti-pituitary and anti-hypothalamus antibodies was simultaneously detected in 52.9% of patients affected by PAH and in no patients carrying a non-secreting pituitary adenoma. As a consequence, although the presence of APAs may not exclude a non-secreting pituitary adenoma, the simultaneous positivity for anti-pituitary and anti-hypothalamus antibodies makes a diagnosis of not-secreting pituitary adenomas unlikely, with an odds ratio of 0.27 (95%IC: 0.13–0.57) [18]. In addition, the detection of APAs positively predicts the outcome of treatment with glucocorticoids in PAH [5].

### 3.2. Putative Antigens of Primary Autoimmune Hypophysitis

Several studies focused on identifying the auto-antigens of PAHs. Lupi et al. [11] demonstrated, through their SJL/J experimental model, that the extracts of whole mouse pituitaries and cytosol fractions had the strongest immunogenic proprieties, with respect to pituitary membranes and nuclei, and that a high immunogen dose is associated with more severe hypophysitis [11]. The immunoblotting of pituitary cytosol proteins and patients’ sera allowed the identification of a 49-kilo Dalton and a 40-kilo Dalton protein respectively in 70% and in 50% of histologically-proven hypophysitis [19]. A subsequent study recognized the 49-kilo Dalton protein as the alpha-enolase [13], which acts as a glycolysis enzyme, a plasminogen receptor, and a controller of cell growth and differentiation, through the downregulation of *c-myc* proto-oncogene expression [20]. Anti-alpha enolase antibodies were detected in other autoimmune diseases, such as mixed cryoglobulinemia, arthritis with kidney involvement, discoid and systemic lupus erythematosus, systemic sclerosis, rheumatoid arthritis, vasculitis with positive anti-neutrophil cytoplasmic antibodies, primary biliary cirrhosis, autoimmune hepatitis, primary sclerosing cholangitis, inflammatory bowel disease, and primary membranous nephropathy [13]. The antibodies anti-GH, anti-pituitary gland specific factor 1a (PGSF1a) and 2 (PGSF2), anti-chorionic somatomammotropin hormone, anti-prohormone convertase, anti-pituitary-specific positive transcription factor 1 (PIT-1), anti-pro-opiomelanocortin (POMC), anti-alpha rad guanine nucleotide dissociation inhibitor (GDI), anti-secretogranin, anti-tudor domain-containing protein (TDRD6) and anti-T-PIT were detected in patients affected by hypophysitis and by hypopituitarism [20,21,22,23,24,25,26,27]. Growth hormone and proopiomelanocortin were also suggested as antigens of IgG4-related hypophysitis [23]. Antibodies against GH, PGSF1a, PGSF2, and T-PIT were also detected in healthy controls and in patients affected by isolated adrenocorticotropic hormone (ACTH) deficit or by other autoimmune diseases [20,21]. Finally, rabphilin-3A was described as a putative antigen of infundibulo-neuro-hypophysitis [28,29].

This different antigenic profile in infundibulo-neuro-hypophysitis may be explained further by the different histological characterization of the adeno-pituitary (which t is composed of epithelial tissue) and of the neuro-pituitary (which is composed of neural tissue). This dual nature of the gland may be due to its embryogenesis. During uterine development, a caudal extension of the primitive forebrain (the diencephalon) grows towards the roof of the primitive oral cavity. Adenohypophysis derives from epithelial cells (the oral ectoderm) and is composed of the pars distalis, a thin layer called the pars tuberalis, and the pars intermedia, which is lost in adult humans. The neurohypophysis develops from a downgrowth of neural tissue at the base of the diencephalon (corresponding to the hypothalamus in the adult) and gives rise to the pars nervosa, the infundibulum, and the median eminence. The infundibulum and the pars tuberalis make up the pituitary stalk [30]. This different embryogenic origin of the adeno-pituitary and the neuro-pituitary may also explain the different inflammatory involvement that occurs in adeno-hypophysitis, infundibulo-neuro-hypophysitis, and pan-hypophysitis. In fact, the inflammatory process involves only the adeno-pituitary in adeno-hypophysitis. It involves the pituitary stalk and the neuro-pituitary in cases of infundibulo-neuro-hypophysitis, and it involves all these structures in cases of pan-hypophysitis [8].

### 3.3. Cell-Mediated Immune Response in Primary Autoimmune Hypophysitis

Few studies have been conducted to investigate the cell-mediated immune response in PAHs. The flow cytometric analysis of experimental hypophysitis conducted on the SIL/J murine models showed that the number of hematopoietic cells increased in immunized mice as compared to not-immunized mice [11]. In fact, the hematopoietic cells accounted for about 85% of all the cells of pituitary extracts in the SIL/J mice and only 2% of the cells of the not-immunized mice The lymphocytes were the most prevalent hematopoietic cells, with CD4-positive T-lymphocytes three times more abundant than CD8-positive T-lymphocytes [11]. The majority of the T-lymphocytes expressed CD44, suggesting an activated/memory phenotype. Furthermore, monocytes/macrophages and granulocytes were detected in SIL/J experimental hypophysitis [11]. In particular, the dendritic CD11-positive cells were identified in close proximity to the lymphocytes aggregate, suggesting that dendritic cells present antigens to infiltrating T-cells, inducing the activation of T-cells and stimulating the secretion of cytokines. The gamma-interferon and the 17 interleukin were detected through cytokine array membranes on pituitary extracts of SIL/J hypophysitis [31]. Moreover, immunohistochemical studies proved the presence of double PCNA/CD3-positive and double PCNA/B220-positive lymphocytes, suggesting that the active T and B-cells proliferated into the pituitary gland [31]. In fact, PCNA is a marker of cell proliferation. This data was also confirmed in a single case report of a female patient with PAH, with a pituitary infiltration of double CD3/Ki67- and double CD20/Ki67-positive lymphocytes [31].

Our recent study confirmed that PAH may be considered an in situ disease. Similarly, we observed that the neutrophil and lymphocyte counts and the neutrophil/lymphocyte ratio were interchangeable in a cohort of 19 cases of PAH and 50 healthy controls [18].

### 3.4. The Genetics of Primary Autoimmune Hypophysitis

Currently, the genetics of hypophysitis are an open issue. The identification of genetic markers for this disease is difficult due to the complex etiology of this disorder and the possible coexistence of other causes of hypopituitarism, such as congenital diseases that lead to an abnormal development of the pituitary gland, as observed in patients who carried the mutations in the *PIT1* or *PROP1* genes [32,33].

Child-onset hypopituitarism is often due to genetic disorders and its identification should take into account the differential diagnosis from pediatric hypophysitis and, more frequently, from pituitary hyperplasia. On the other hand, there are relevant data about the genetic components underlying autoimmune disorders. In PAH, the same HLA polymorphisms were described.

Beressi et al. [34] reviewed 17 PAH cases that featured the HLA genotype from 1987 to 1999 and described the presence of HLA-DR4 in 44% of cases and of HLA-DR5 in 23% of cases. Instead, more recently, in a series of 15 PAH patients, the HLA haplotypes DQ8 and DR53 were identified, respectively, in 87% and 80% of cases [35].

In a recent study, we identified the presence of 12 HLA haplotypes associated with celiac disease in 16 consecutive Caucasian patients affected by PAH. In particular, the prevalence of the DQ8 haplotype was 25%, a result that was significantly higher than that observed in a control group of 250 consecutive Caucasian individuals (7.2%) [36]. Moreover, the individuals carrying the DQ8 haplotype were four times more likely to develop PAH. The presence of the DQ8 haplotype represents a susceptibility for the occurrence of hypophysitis, as was also observed for celiac disease [36]. In fact, in celiac disease, the heterodimer DQ8 is produced by the combination of two alleles, DQA1*03 and DQB1*0302, which often segregate with the DR4, as the segregation of these two alleles occurs en-bloc. The presence of these alleles in the population determines a higher risk for the development of celiac disease [37]. When the DQ8 is generated by the B1*02 allele (also called “positive B1*02”), it is associated with a risk of celiac disease of around 1:24 in the general population. Instead, the presence of the serological DQ8 (called “B1*02 negative”) is not determined by the B1*02 allele and is associated with a lower risk of celiac disease onset (1:89) [37].

The presence of these alleles is a necessary but not a sufficient factor for the development of PAH and celiac disease. There are many other factors, currently unknown, that can contribute to the onset of these disorders. Therefore, HLA-DQ8 molecular testing may be included in the diagnostic flow-chart of PAH, to help the physician in the differential diagnosis of focal pituitary lesions and non-secreting pituitary adenomas. The HLA system has a modular organization with some genetic loci (beta-DQ, alpha-DQ, DRB1, DRB2, DRA) that are all close to each other. For this reason, it is very common for an allele in DQA or DQB to be co-segregated with an allele in the DRB locus. The HLA system is a gene cluster located on the sixth chromosome and includes more than 200 coding genes, encoding cell-surface proteins responsible for the regulation of the immune system. In humans, three classes of HLA were identified: HLA class I genes (A, B, C), HLA class II genes (DP, DQ, DR) and HLA class III genes (those of the complement factors C2 and C4, and TNF) [38]. Class II HLA alleles are pronominally involved in autoimmune disease. HLA is made up of more than 7000 different alleles, and about 2000 of these alleles belong to the DR gene. These alleles encode for 891 different proteins and 22 null alleles (alleles that do not produce any type of protein) [38]. Beta-DQ has about 165 alleles, and their combination produces about 116 different proteins. Putting together all the possible mixtures that can be generated by the presence of the different HLA alleles, there are about 9 billion different possible combinations. Given this great heterogeneity, HLA is able to respond to a large number of antigens and exogenous substances. In the genetic asset of any subject, different maternal and paternal alleles are combined together for the different genetic loci (DRB, HLA-B, HLA-A, etc.). From this combination, different major histocompatibility complexes (MHCs) are formed and encode glycoproteins, which are expressed only on macrophages, B-cells, and dendritic cells. Their main function is to present antigens to competent cells (i.e., T-helper lymphocytes), determining the immunological response [38].

Therefore, the HLA system is crucial in the immune response, and its dysfunction is associated with a whole range of autoimmune diseases.

### 3.5. Molecular Mechanisms in Immunotherapy Induced Hypophysitis

Hypophysitis is a clinically significant endocrine toxicity in patients receiving treatment with immune check-point inhibitors (ICIs), such as monoclonal antibodies (mAbs), against the cytotoxic T-lymphocyte antigen 4 (CTLA4) and programmed cell death-1 (PD-1) [1]. However, few data and evidence are provided on the molecular mechanism of this disorder.

CTLA-4 is expressed on the cell surface of active CD4-positive and CD8-positive T-cells and binds the CD80 and CD86 that are expressed on the cell surface of antigen-presenting cells (APCs), with higher affinity and avidity than CD28 [39]. The engagement of CTLA-4 with CD80 and with CD86 mitigates the immune response [40]. The primary activity of the CTLA-4 pathway is the draining of the lymph nodes, where naïve T-cells are primed by exposure to tumor antigens (presented by APCs) and become activated [41]. The mAbs act through the inhibition of the CTLA-4 pathway, with a subsequent over-activation of T-lymphocytes that may predispose to the onset of autoimmune disease, as IIH.

Similarly, PD-1 acts as a second inhibitory receptor and is expressed mainly on activated CD8-positive T-lymphocytes [42]. PD-1 is triggered by PD-ligands 1 and 2 (PD-L1 and PD-L2, respectively), which constitutively reside on tumor cells [43,44]. The sites of activation of the PD-1 pathway are the peripheral tumoral tissues and the tumor microenvironment (TME) [41]. The binding of PD-1/PD-L1 suppresses the activity of T-cells [45], converts T-helpers into T-regulatory cells [46] and activates pro-survival signaling pathways in cancer cells through a mechanism of resistance to cytotoxic T-lymphocytes [47]. Treatment with mAbs anti-PD-1 and with anti-PD1L is associated with a high frequency of autoimmune disorders, also involving endocrine toxicity [1].

The genetic polymorphisms of the *CTLA-4* and of *PD-1* genes can increase the incidence of autoimmune disease, including IIH [48,49]. Many of these polymorphisms do not actually change the CTLA-4 amino acid sequence, but can modify the affinity for the CTLA-4 mAb, increasing the risk of occurrence of immunotherapy-induced autoimmune disorders [48]. In the diagnostic work-up, there is no recommendation to investigate the *CTLA-4* gene polymorphisms. Among the genetic mechanisms behind the IIH, this mechanism should also be taken into consideration.

Cases of isolated ACTH deficiencies are also observed during immunotherapy, in the absence of the typical radiological features of hypophysitis. Isolated ACTH deficiencies are mainly classified as congenital or acquired. In congenital forms, the two most commonly involved genes are the *T-PIT* gene (better known as the TBX19 gene) and the *POMC* gene (the *melanocortin* gene) [50]. According to more recent research, IIH and ACTH deficits may be considered two different endocrine toxicities. In a recent study, conducted on 62 cancer patients treated with ICIs, the prevalence of APAs was similar among the five patients who developed an IIH (APA positivity: 80%) and in those who developed an ACTH deficit (APA positivity: 88.2%) [51]. Moreover, Kanie at al. [52] recently described two patients who developed anti-ACTH antibodies and central hypoadrenalism during ICI therapy. The authors observed ACTH expression on the tumor cell surface of a kidney neoplasia and of a melanoma, and suggested that the ectopic expression of cancer cells in ACTH can promote the immune response with the synthesis of anti-ACTH antibodies that can also act on the pituitary corticotroph cells, inducing central hypoadrenalism. In this view, the ACTH deficit may be considered as a paraneoplastic syndrome.

Some studies also investigated HLA haplotypes in patients affected by IIH and immunotherapy-induced hypopituitarism, showing that the prevalence of HLA-Cw12 and HLA-DR15 was significantly higher in patients with immunotherapy-induced hypophysitis. Instead, the prevalence of HLA-Cw12, HLA-DR15, HLA-DQ7, and HLA-DPw9 was significantly higher in patients with immunotherapy-induced central hypoadrenalism [51]. In a study conducted on a Japanese population of patients with pituitary disfunction during ICIs, HLA-DR15, B52, and Cw12 were identified as possible predisposing factors [53]. The putative antigens of immunotherapy-induced hypophysitis and hypopituitarism were not completely clarified, despite preliminary studies suggesting ACTH, prolactin, and CTLA-4 as possible disease antigens [54,55].

## 4. Clinical Application of Molecular Biomarkers in Hypophysitis

The diagnosis of hypophysitis is performed through clinical criteria in the large majority of cases, as a certain diagnosis is possible only through the pathological examination of the pituitary gland [6].

Hypophysitis should be suspected in patients with symptoms of hypopituitarism and/or neurological signs. Contrasted magnetic resonance images (MRI) of the diencephalon-pituitary region play a crucial role in the diagnosis of hypophysitis through the identification of the typical radiological signs (enlargement of the pituitary gland, swelling of the pituitary stalk and loss of neuro-pituitary signal in the T1-weighed images) and through the exclusion of pituitary gland and stalk focal lesions (such as non-secreting pituitary adenoma, craniopharyngioma, germinoma, meningioma, glioma, and pituitary and infundibular metastasis). The differential diagnosis between hypophysitis and focal lesions may represent a great clinical challenge.

With this in mind, molecular and genetic biomarkers can be integrated into the diagnosis. In Figure 1 and Figure 2, we summarize our proposed flow-charts for the diagnosis of primary and immunotherapy-induced hypophysitis.

Autoimmune hypophysitis should be suspected in patients with pituitary dysfunction or neurological/ophthalmological disorders (such as headaches and visual field defects) associated with the typical neuroradiological features [2]. After ruling out focal hypothalamic-pituitary lesions and secondary causes of hypophysitis, APA identification and HLA genotyping may be useful for reinforcing the diagnosis of clinically suspected PAH or IIH. Currently, guidelines for the therapeutic management of patients affected by PAH and IIH are not available. According to our clinical practice, in cases of PAH, immunosuppressive treatment with glucocorticoids, azathioprine, or mofetil mycophenolate should be recommended in young patients and in those affected by partial or complete hypopituitarism, diabetes insipidus, or neurological or ophthalmological symptoms, after ruling out the major contraindications to the treatments [5]. In fact, the recovery of the pituitary function after immune-suppressive treatments may be reached in a non-negligible percentage of cases, in particular for central hypogonadism and growth hormone deficit [5]. A strict follow-up should be performed both for PAH patients treated with immune-suppressive therapies and for patients conservatively managed with clinical and radiological observation. In fact, according to our clinical practice, a clinical, endocrine and radiological evaluation should be conducted within at least three months of the diagnosis of PAH or of the prescription of immunosuppressive treatment. As PAH typically responds to immunosuppressive therapies, the worsening of clinical and morphological features necessarily suggests alternative pathologies with a hypophysitis-like picture, such as germinomas or Langerhans cell histiocytosis, which represents a strong differential diagnosis, especially in children and adolescents. In fact, PAH is rare in the pediatric population, as the mean age at the onset of the disease is 34.5 years for women and 44.7 years for men [56]. Instead, intracranial germinoma are diagnosed in around two-thirds of cases below 18 years of age, highlighting the preponderance of this disease in the first two decades of life [57]. Similarly, Langerhans cell histiocytosis is more frequently encountered in children aged 0–15 years, with an estimated annual incidence of up to 8.9 cases per million in children younger than 15 years and a male-to-female ratio of 2:1 [58]. In these patients, cerebrospinal fluid examination may be useful for the identification of neoplastic and inflammatory cells [59], but also to rule out infective hypophysitis.

For patients with IIH, immunosuppressive therapy with corticosteroids should be reserved only for cases with neurological involvement, in order to guarantee the efficacy of ICI treatment [1]. Furthermore, for IIH patients, a strict follow-up is recommended, which includes conducting hormonal tests and pituitary MRI within a month of IIH diagnosis. In fact, the radiological signs of hypophysitis typically reappear within a short time. Moreover, as IIH appears as adeno-hypophysitis during follow-up, pituitary metastasis should be considered as an alternative diagnosis, in case of the persistent enlargement of the pituitary gland. In Figure 3, typical pictures of pituitary MRIs of adeno-hypophysitis, infundibulo-neuro-hypophysitis, pan-hypophysitis are presented. Instead, in Figure 4, we reported same clinical cases of patients affected by hypothalamus-pituitary diseases that should be considered in the differential diagnosis of hypophysitis, as multifocal germinoma, Langerhans cell histiocytosis and pituitary stalk metastasis of a breast cancer.

## 5. Conclusions

Hypophysitis is a very complex pathological condition. Its diagnosis, treatment, and follow-up require a multidisciplinary team. New knowledge about the genetic and molecular pathways of hypophysitis may be of great importance for the diagnosis of hypophysitis. After ruling out focal pituitary lesions, the study of anti-pituitary antibodies and HLA genotyping should be integrated in the diagnostic steps for reinforcing the clinical diagnosis of autoimmune disorder of the pituitary gland. In this view, the presence of a multidisciplinary team in healthcare facilities becomes increasingly essential, especially in the context of collaboration with physicians and molecular diagnostics laboratories.

## Figures and Tables

**Figure 1 jpm-11-01026-f001:**
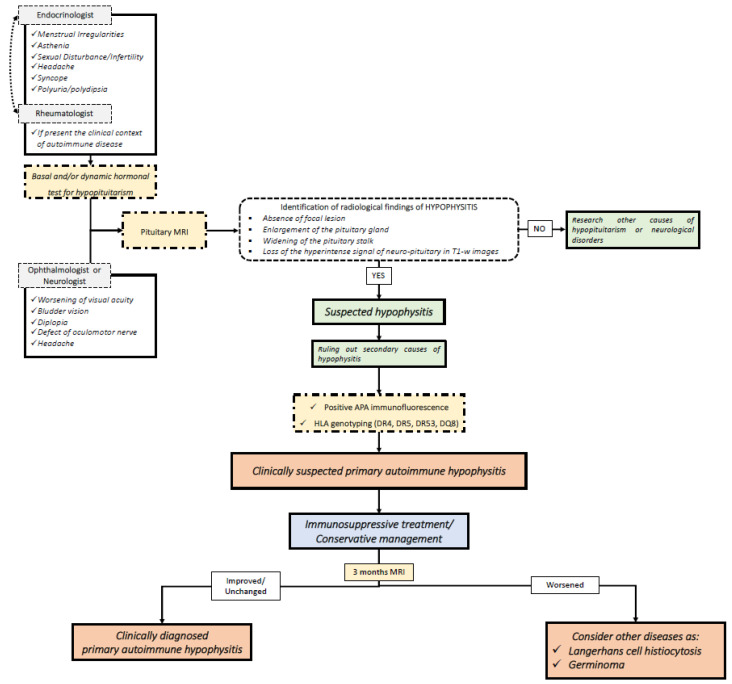
Flow-chart for the diagnosis of primary autoimmune hypophysitis. MRI: magnetic resonance imaging; APA: anti-pituitary antibodies; HLA: human leukocyte antigen.

**Figure 2 jpm-11-01026-f002:**
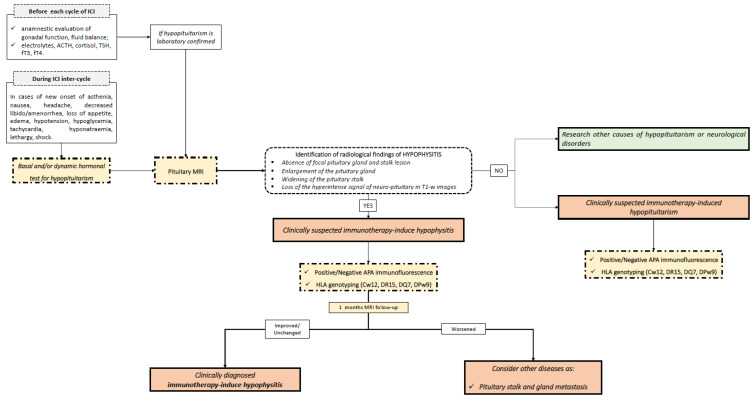
Flow-chart for the diagnosis of immunotherapy-induced hypophysitis and immunotherapy-induced hypopituitarism. ACTH: adrenocorticotropic hormone; TSH: thyroid stimulating hormone; fT3: free triiodothyronine; fT4: free triiodothyronine; APA: anti-pituitary antibodies; HLA: human leukocyte antigen.

**Figure 3 jpm-11-01026-f003:**
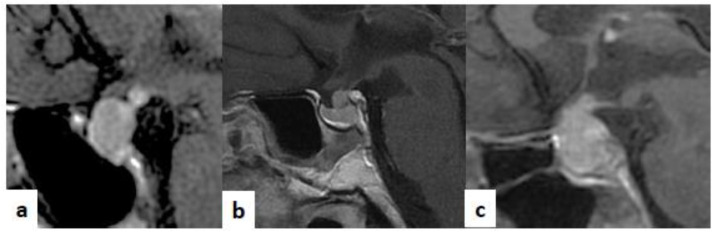
Sagittal T1-weighted (T1-w) images after contrast medium injection. (**a**) A case of adeno-hypophysitis. The adeno-pituitary and the pituitary stalk are swollen. The hyper-intense signal of the neuro-pituitary is normally present on the T1-w image. (**b**) A case of infundibulo-neuro-hypophysitis. The pituitary gland is reduced in size. The pituitary stalk is swollen. The hyper-intense signal of the neuro-pituitary is lost on the T1-w image. (**c**): A case of pan-hypophysitis. The pituitary gland and stalk are swollen. The hyper-intense signal of the neuro-pituitary is lost on the on T1-w image.

**Figure 4 jpm-11-01026-f004:**
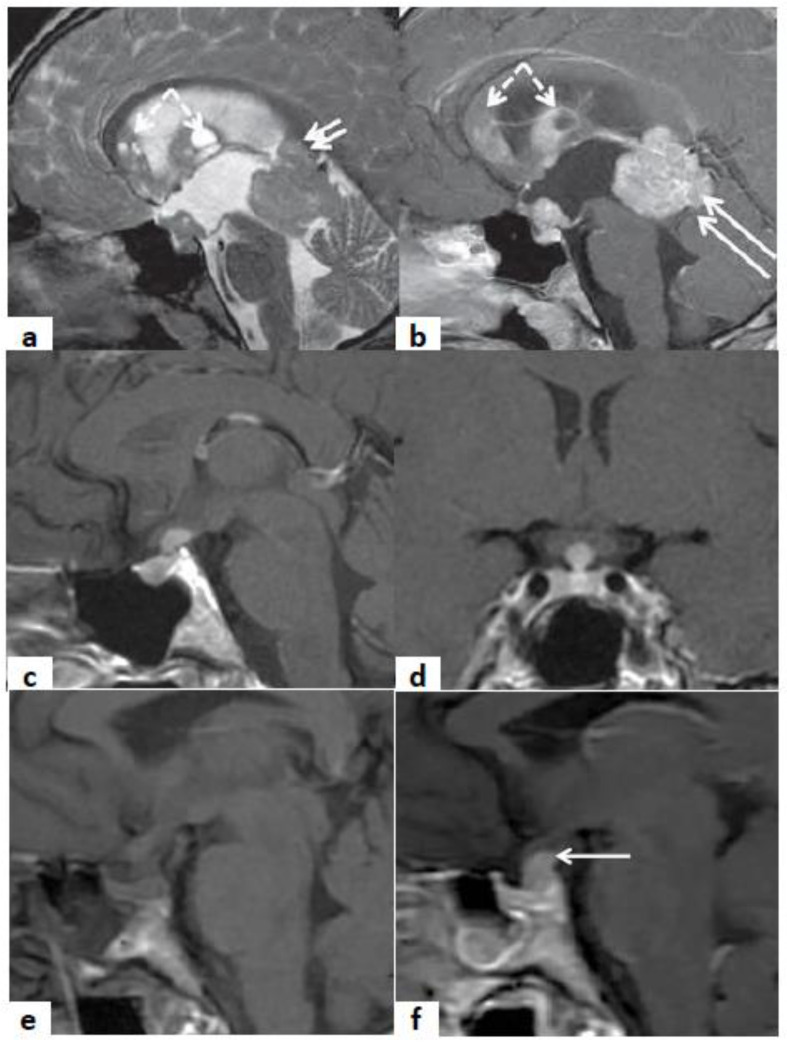
Sagittal T2w (**a**) and CE-T1w (**b**) MR images. A case of a 20-year-old male patient affected by multifocal germinoma. The MR images show a pathologic tissue that involves the anterior portion of the third ventricle and the pineal gland (arrows). The multiple ependymal contrast-enhanced nodules (frecce tratteggiate) are suggestive of metastatic dissemination (dashed arrows). Sagittal (**c**) and coronal (**d**) MR CE-T1w images. A case of an 18-year-old female patient affected by Langerhans cell histiocytosis. The MR images show a contrast-enhanced pathologic tissue that involves III ventricle infundibulum. The pituitary gland shows a normal size. Sagittal MR T1w (**e**) and CE-T1w images (**f**). A case of a 50-year-old female with III ventricle infundibulum and pituitary stalk metastasis of breast cancer. The MR images show a pathologic contrast-enhanced tissue that involves III ventricle infundibulum and the pituitary stalk (arrow in (**b**)). The hyper-intense signal of the neuro-pituitary is lost on the T1-w image.

**Table 1 jpm-11-01026-t001:** Etiological classification of hypophysitis.

Etiological classification	Primary Hypophysitis	The Inflammatory Process Involves Only the Pituitary Gland	
	Secondary hypophysitis	Cystic/neoplastic pituitary lesions	Craniopharyngioma, adenoma, germinoma, cyst and Rathke cleft cyst
		Autoimmune systemic diseases	IgG4-disease, sarcoidosis, granulomatosis, vasculitis, Crohn’s disease
		Histiocytic proliferative diseases	Langerhans cell histiocytosis, Erdheim–Chester, Rosai–Dorfman diseases
		Infective Systemic Disease	Syphilis, tuberculosis, fungal and viral infection in immune-depressed patients
	Immuno-therapy induced hypophysitis	in patients on treatment with check-point inhibitors, as monoclonal antibodies anti-CTLA-4, anti-PD-1 and anti-PD1 ligand	

**Table 2 jpm-11-01026-t002:** Biomarkers in primary autoimmune and immune-therapy induced hypophysitis, hypopituitarism and ACTH deficit.

Biomarkers		Primary Autoimmune Hypophysitis	Immunotherapy-Induced Hypophysitis	Immunotherapy-Induced Hypopituitarism/ACTH Deficit
Humoral immune response	Anti-pituitary antibodies	Yes	Yes	Yes
	Putative antigens	Alpha-enolase Chorionic somatomammotropin hormone Growth hormone (typically in AH) Prohormone convertase PC2 regulatory protein PGSF 1a and 2 (typically in AH and INH) Pit-1 POMC Alpha-rab guanine nucleotide dissociation inhibitor Rabphilin-3A (typically in INH) Secretogranin II T-PIT Tudor Domain Containing Protein 6	ACTH CTLA-4 Prolactin	ACTH
	Cytokine	Gamma-interferon 17 interleukin	No data	No data
Cell-mediated immune response	Lymphocytes	B220, CD3, CD4, CD8, CD20, CD44 positive	No data	No data
	Other	CD11 positive dendritic cells monocytes/macrophages granulocytes	No data	No data
Genetic biomarkers	HLA haplotypes	DR4, DR5, DR53, DQ8	Cw12, DR15	Cw12, DR15, DQ7, DPw9
	Other	No data	*CTLA-4* and *PD-1* gene	No data

## Data Availability

Not applicable.

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
