# Peer review of "Molecular and Genetic Immune Biomarkers of Primary and Immune-Therapy Induced Hypophysitis: From Laboratories to the Clinical Practice"

_jpm, 2021, doi:10.3390/jpm11101026_

Round 1

Reviewer 1 Report

The authors review the evidence around humoural, cellular and genetic biomarkers of hypophyitis, and suggested an algorithm for distinguishing hypophysitis from focal pituitary lesions and differnetiating different causes of hypophysitis.

However there are numerous statements throughout the text that are either not expanded on or seem incorrect. For example in line 54 mention is made of an increase in the number of reported cases but the reasons for this are not discussed. In line 66 it is stated that the large majority of non-functioning pituitary tumours require surgery but this is not the case.

Table 1 and the classification is useful

In Table 2 no definition of the sub-classification of Immunotherapy induced hypophysitis is given, and the table contradicts  the text in places eg line 179.

In line 295 the authors make comment of polymorphisms in CTLA-4 that do not change the amino acid sequence but modify the affinity of the mAB. No reference is given or explanation of possible mechanism for this effect.

In line 308 I am unclear about the justification for describing a toxicity of treatment as a paraneoplastic effect.

Whilst the flowcharts for diagnosis are interesting and may be of use, they do not integrate any of the biomarkers discussed elsewhere in the paper, and hence do not really seem an advance over current practice as described in the introduction. 

Author Response

The authors thank the Reviewer for him/her comments on the manuscript, that allowed as to improve the quality of our paper. 

Point 1) "...there are numerous statements throughout the text that are either not expanded on or seem incorrect. For example in line 54 mention is made of an increase in the number of reported cases but the reasons for this are not discussed. In line 66 it is stated that the large majority of non-functioning pituitary tumours require surgery but this is not the case..."

We provided to better detailed same statements in the text. In particular we discussed the advances in the diagnosis of hypophysitis and we better detailed indication the removal of pituitary lesions. 

2) Table 1 and the classification is useful

We thank for this comment

3) In Table 2 no definition of the sub-classification of Immunotherapy induced hypophysitis is given, and the table contradicts  the text in places eg line 179.

In table 2, we better defined the antigens according to subtypes of hypophysitis. However, we can't understand  what the Reviewer means in term of sub-classification of immunotherapy induced hypophysitis. Moreover, the line 179 in the text in not referred to table 2, problably for the different formatting of the files. If the reviewer will be help us in identifying the question we will be happy to better answer to the question. Thank you. 

4) In line 295 the authors make comment of polymorphisms in CTLA-4 that do not change the amino acid sequence but modify the affinity of the mAB. No reference is given or explanation of possible mechanism for this effect.

Thank you. We add the reference. 

5) In line 308 I am unclear about the justification for describing a toxicity of treatment as a paraneoplastic effect.

We thank the reviewer for her/him comment. We better described the possible paraneoplastic effect as reported by Kanie, K.; Iguchi, G.; Bando, H.; Urai, S.; Shichi, H.; Fujita, Y.; Matsumoto, R.; Suda, K.; Yamamoto, M.; Fukuoka, H.; Ogawa, W.; Takahashi, Y. Mechanistic Insights into Immune Checkpoint Inhibitor-Related Hypophysitis: A Form of Paraneoplastic Syndrome. Cancer Immunol. Immunother. 2021. https://doi.org/10.1007/s00262-021-02955-y.

6) Whilst the flowcharts for diagnosis are interesting and may be of use, they do not integrate any of the biomarkers discussed elsewhere in the paper, and hence do not really seem an advance over current practice as described in the introduction. 

We thank the Reviewer again for him/her comment. We reviewed the flowcharts adding possible suggestions for integrate the molecular biomarkers in the diagnosis of hypophysitis. 

Reviewer 2 Report

Thank you for the opportunity to review the manuscript “Molecular and genetic immune biomarkers of primary and immune-therapy induced hypophysitis: from the laboratories to the clinical practice.” by Chiloiro Sabrina et al. The diagnosis and treatment of hypophysitis is very challenging for clinicians. Therefore, I find the paper interesting and valuable for clinical practice. 

However, there are some points that need to be improved.

  1. Abstract, lines 46-47 “The diagnosis of hypophysitis is complex, because is based on clinical criteria.” I would add “clinical and radiological”, as the Authors mention later in the text: lines 61-62 “The identification of the typical radiological findings of hypophysitis play a crucial role in the diagnosis.”
  2. Lines 72-73 “For these reason a definitive diagnosis of hypophysitis based on univocal clinical is advocated.”- unclear
  3. Line 64 “immunosuppressive treatment or may be conservative managed”- please specify
  4. Lines 77-81 lack of references
  5. Lines 81-83 “At least, in the more recent years, hypophysitis has recognized as a clinically significant endocrine toxicity in patients on treatment with immune check point inhibitors.”- please give more information about the ICI treatment.
  6. Line 124 and following- lack of information that APA may be found also in healthy subjects and that the presence of APA does not exclude neoplastic lesions.
  7. Fig 1- the last part belongs to Fig2?
  8. Fig 1 and 2- abbreviations should be explained.
  9. In my opinion Fig 1 and 2 should be modified. The MRI findings are not always explicit, as the Authors explained in the paper, therefore the diagnostic path is not so simple. The same MRI findings may be present in hypophysitis as well as in germinoma. Moreover, there is lack of recommendations for control MRI scans- how often should they be performed?
  10. I would suggest the usage of “widening” rather than “swelling of the pituitary stalk” (Fig 1 and 2, line 325).
  11. There is little information in the manuscript of age and sex as potential risk factors for different types of diagnosis, e.g. stalk widening in adolescence is highly predictive of germinoma.
  12. Please explain what is a place for the cerebrospinal fluid examination in the diagnostic process?
  13. I would strongly recommend adding MRI pictures most typical in hypophysitis and in other differential diagnoses.
  14. I think the Authors could drive more conclusions, e.g. what are the most reliable autoimmune or genetic tests that may be used as a first line examinations in clinical practice.
  15. Lack of: Author Contributions, Conflicts of Interest etc.

Author Response

We thank the reviewer for him/her comments, that allowed us to improve the quality of our paper.

Abstract, lines 46-47 “The diagnosis of hypophysitis is complex, because is based on clinical criteria.” I would add “clinical and radiological”, as the Authors mention later in the text: lines 61-62 “The identification of the typical radiological findings of hypophysitis play a crucial role in the diagnosis.”

We thank the reviewer for her/him comment and add the sentence at line 47 page 2 in abstract

Lines 72-73 “For these reason a definitive diagnosis of hypophysitis based on univocal clinical is advocated.”- unclear

We thank the reviewer for her/him comment and we modified the sentence at line 77-78 page 3

 Line 64 “immunosuppressive treatment or may be conservative managed”- please specify

We thank the reviewer for her/him comment and we modified the sentence at line 70-73 page 3

Lines 77-81 lack of references

We thank the reviewer for her/him comment and we provided to add references (reference number: 7)

Lines 81-83 “At least, in the more recent years, hypophysitis has recognized as a clinically significant endocrine toxicity in patients on treatment with immune check point inhibitors.”- please give more information about the ICI treatment.

We thank the reviewer for her/him comment and we provided to add more information about ICI, at line 89-95 page 4.

Line 124 and following- lack of information that APA may be found also in healthy subjects and that the presence of APA does not exclude neoplastic lesions.

We thank the reviewer for her/him comment and we provided to better detailed as required at line 149 page 6 and at line 153-160 page 6.

  1. Fig 1- the last part belongs to Fig2? Fig 1 and 2- abbreviations should be explained. In my opinion Fig 1 and 2 should be modified. The MRI findings are not always explicit, as the Authors explained in the paper, therefore the diagnostic path is not so simple. The same MRI findings may be present in hypophysitis as well as in germinoma. Moreover, there is lack of recommendations for control MRI scans- how often should they be performed?

We reviewed the figures and detailed the abbreviations in captions 

  1. I would suggest the usage of “widening” rather than “swelling of the pituitary stalk” (Fig 1 and 2, line 325).

We thank the reviewer for him/her comment and we modified accordingly

  1. There is little information in the manuscript of age and sex as potential risk factors for different types of diagnosis, e.g. stalk widening in adolescence is highly predictive of germinoma. Please explain what is a place for the cerebrospinal fluid examination in the diagnostic process?

We thank the reviewer for him/her comment and we extended this section at line 364-372 pages 12-13.

  1. I would strongly recommend adding MRI pictures most typical in hypophysitis and in other differential diagnoses.

We added the pictures.

  1. I think the Authors could drive more conclusions, e.g. what are the most reliable autoimmune or genetic tests that may be used as a first line examinations in clinical practice.

We thank the reviewer for him/her comment and we extended this section

  1. Lack of: Author Contributions, Conflicts of Interest etc.

We thank the reviewer for him/her comment and we extended this section

Round 2

Reviewer 2 Report

Please see the attached review.
